# PARTIAL ADVANTAGE ESTIMATOR FOR PROXIMAL POLICY OPTIMIZATION

## ABSTRACT

Estimation of value in policy gradient methods is a fundamental problem. Generalized Advantage Estimation (GAE) is an exponentially-weighted estimator of an advantage function similar to $\lambda$-return. It substantially reduces the variance of policy gradient estimates at the expense of bias. In practical applications, a truncated GAE is used due to the incompleteness of the trajectory, which results in a large bias during estimation. To address this challenge, instead of using the entire truncated GAE, we propose to take a part of it when calculating updates, which significantly reduces the bias resulting from the incomplete trajectory. We perform experiments in MuJoCo and $\mu$RTS to investigate the effect of different partial coefficient and sampling lengths. We show that our partial GAE approach yields better empirical results in both environments.

## 1 INTRODUCTION

In reinforcement learning, an agent judges its state according to its environment, then selects an action, and iterates these steps, constantly learning from the environment. Let $S$ be the set of states and $A$ the set of actions. The selection process is quantified by the transition probability $P$. When the agent makes the choice to execute an action, it will receive feedback rewards from the environment, that is, $r : S \times A \rightarrow [R_{min}, R_{max}]$. In this way, the model will produce a trajectory sequence $\tau$, that is $\tau : (s_1, a_1, ..., s_T, a_T)$. This trajectory will have a cumulative return, $\sum_{t=1}^{T} \gamma^{t-1} r_t$, where $\gamma$ is the discount factor and $T$ is the number of steps performed. The goal of reinforcement learning is to find an optimal policy $\pi$, so that an agent can obtain the maximum cumulative expected cumulative reward using this policy. Policy refers to the probability of specifying an action in each state, $\pi(a|s) = p(A_t = a|S_t = s)$. Under this policy, the cumulative return follows a distribution, and the expected value of the cumulative return at state $s$ is defined as a state-value function:

$$v_\pi(s) = E_\pi[\sum_{k=0}^{\infty} \gamma^k r_{t+k+1} | S_t = s] \tag{1}$$

Accordingly, at state $s$, the expected value of cumulative return after executing action $a$ is defined as a state action value function:

$$q_\pi(s) = E_\pi[\sum_{k=0}^{\infty} \gamma^k r_{t+k+1} | S_t = s, A_t = a] \tag{2}$$

Often the temporal difference (TD) algorithm or a Monte Carlo (MC) method is used to estimate the value function. However, each of these methods has its advantages and disadvantages. The TD algorithm value estimator has the characteristics of high bias and low variance. In contrast, the estimator of the MC algorithm has low bias and high variance. Kimura et al. (1998) put forward a method to skillfully find a balance between bias and variance, which is called $\lambda$ return. TD($\lambda$) proposed by Sutton (1988) is a variant of the $\lambda$ return that provides a more balanced value estimation.

Generalized Advantage Estimation (GAE) is a method proposed by Schulman et al. (2015b) to estimate the advantage value function. In fact, it is a method to apply the lambda return method to estimate the advantage function. As proposed in the PPO paper Schulman et al. (2017), in practical applications, due to the incompleteness of the trajectory, truncated GAE is used, which leads to large bias in the estimation process. For this, we propose *partial* GAE that uses partially calculated GAEs, rather than the entire truncated GAEs, to significantly reduce the bias that caused

by incomplete trajectories. In addition to experiments in common MuJoCo environments, we also conduct experiments in the complex and challenging $\mu$RTS environment. Our methods have been empirically successful in these environments.

## 2 VALUE ESTIMATOR

The gradient in the policy gradient algorithm is generally written in the following form:

$$g = \mathbb{E}\left[\sum_{t=0}^{\infty} \Psi_t \nabla_\theta log\pi_\theta(a_t|s_t)\right] \tag{3}$$

where $\Psi_t$ is used to control the update amplitude of the the policy update in the gradient direction. In the basic policy gradient algorithm, the action value function $q_\pi(s,a)$ is used as $\Psi_t$, and $q_\pi(s,a)$ is estimated by cumulative return $G_t$. One of the most direct methods of estimating the value function is the Monte Carlo (MC) method. MC starts directly from the definition of the value function, and takes the accumulation of return values as the estimator of the value function. The accumulation of the discounted reward $\sum_{n=0}^{N} \gamma^n R_{t+n}$ of a reward sequence $(r_t, r_{t+1}, ..., r_{t+N})$ is taken as the estimator of the state value under state $s_t$. The state value estimator of the MC algorithm is unbiased estimator. In addition, because the random variables in the estimator are all the returns after time $t$, and the dimensions are high, the estimator has the characteristic of high variance.

The TD algorithm uses $r_t + \gamma V_\theta(s_{t+1})$ as the estimator of value function $V^\pi(s_t)$. There is a certain error, denoted as $e_\theta$, between the approximate value function and the real value function. Equation 4 can be obtained, and the bias $\gamma E_{S_{t+1}}[e_\theta(S_{t+1})]$ can be obtained by using the TD algorithm. In addition, because there are fewer dimensions of random variables in the estimator, the variance of the estimator remains low.

$$E_{(r_t, S_{t+1})}[r_t + \gamma V_\theta(S_{t+1})] = V_{S_t}^\pi + \gamma E_{S_{t+1}}[e_\theta(S_{t+1})] \tag{4}$$

Compared with the TD and MC methods, the $\lambda$-return method seeks to find a balance between bias and variance. In Equation 6, if $\lambda$ is 0, it is the estimator of the TD method, and if $\lambda$ is 1, it is the estimator of the MC method.

$$G_t^{(n)} = \gamma^n V(s_{t+n}) + \sum_{l=0}^{n-1} \gamma^l r_{t+l} \tag{5}$$

$$G_t^\lambda = \lambda^{N-1} G_t^{(N)} + (1-\lambda) \sum_{n=1}^{N-1} \lambda^{n-1} G_t^{(n)} \tag{6}$$

One disadvantage of using $q_\pi(s,a)$ is that has a large variance, which can be reduced by introducing the baseline $b(t)$,

$$g = \mathbb{E}\left[\sum_{t=0}^{\infty} (q_\pi(s,a) - b(t))\nabla_\theta log\pi_\theta(a_t|s_t)\right] \tag{7}$$

Using the average incomes from the sample as a baseline offers some improvement. However, for the Markov process, the baseline should change according to the state, and the state baseline which has large value for all actions should be larger, and vice versa. While the A2C and A3C algorithms use $V_t$ as the baseline, $q_\pi(s,a) - V_t$ is called the advantage, $Q(S,A)$ can be approximated as $R + \gamma V'$, and the advantage is $A(s_t, a_t) = r_t + \gamma V(s_{t+1}) - V(s_t)$. In the Generalized Advantage (GAE) paper Schulman et al. (2015b), a TD($\lambda$)-like method is proposed, in which the weighted average of the estimated values of different lengths provides the estimated value. This method to calculate the advantage is adopted by powerful reinforcement learning algorithms such as TRPO Schulman et al. (2015a) and PPO Schulman et al. (2017).

$$\delta_{t+l}^V = r_{t+l} + \gamma V(s_{t+l+1}) - V(s_{t+l}) \tag{8}$$

$$\hat{A}_t^{GAE(\gamma,\lambda)} = \sum_{l=0}^{\infty} (\gamma\lambda)^l \delta_{t+l}^V \tag{9}$$

When the parameter $\gamma$ is introduced to estimate the policy gradient, $g^\gamma$ is biased from $g$. Such a policy gradient estimation problem actually aims at discounted cumulative reward. In the later part of this paper, the GAE is mainly discussed.

In addition to the above method for calculating $\Psi_t$, there are different methods for estimating the value function Bertsekas et al. (2011). In general, the value loss $L_t^{VF}$ is calculated with squared-error $(V_{\theta_t} - V_t^{target})^2$. The GAE paper Schulman et al. (2015b) uses the trust region method to optimize the value function in each iteration of the batch optimization process.

$$\begin{aligned} \underset{\phi}{minimize} \quad & \sum_{n=1}^N \| V_\phi(s_n) - \hat{V}_n \| \\ \text{subject to} \quad & \frac{1}{N} \sum_{n=1}^N \frac{\|V_\phi(s_n) - V_{\phi_{old}}(s_n)\|}{2\sigma} \leq \epsilon \end{aligned} \tag{10}$$

And, in most implementations, $V_{\theta_t}$ is clipped around the value estimates on both sides $V_{\theta_{t-1}}$ and $V_{\theta_{t+1}}$.

$$L_t^{VF} = max[(V_{\theta_t} - V_t^{target})^2, clip(V_{\theta_t}, (V_{\theta_{t-1}} + \epsilon, V_{\theta_{t+1}} + \epsilon) - V_t^{target})^2] \tag{11}$$

The work of Tucker et al. (2018) proposes normalizing the advantage, which is shown to improve the performance of the policy gradient algorithm. After the GAE calculates advantages in a batch, the mean and standard deviation of are computed. Then for each advantage, one subtracts the mean and divides by the standard deviation. In Eq. 12, $A_i^{norm}$ is normalized advantage, $A_i$ is advantage, $A^{mean}$ is mean of advantages, $A^{std}$ is standard deviation of advantages

$$A_i^{norm} = \frac{A_i - A^{mean}}{A^{std}} \tag{12}$$

It is very important for the policy gradient algorithm to estimate a more instructive value function. In the following section, we will mainly discuss the practical application of GAE and how it can be improved.

## 3 PARTIAL GAE

In the actual environment, a task often has terminal states, which result in a finite trajectory length. For the trajectory terminus at time $D$, one performs an iterative calculation from back to front to compute the GAE. Denote complete trajectory GAE $\hat{A}_t^{GAE(\gamma,\lambda,D)}$ as Eq. 13. $\hat{A}_t^{GAE(\gamma,\lambda,\infty)}$ can be generalized as $\hat{A}_t^{GAE(\gamma,\lambda,D)}$, when $D \to \infty$.

$$\hat{A}_t^{GAE(\gamma,\lambda,D)} = \sum_{l=0}^{D-t} (\gamma\lambda)^l \delta_{t+l}^V \tag{13}$$

In practical applications, the length of one sample is fixed in order to carry out parallel computing more efficiently. In another case, it takes a long time to sample a complete trajectory. In order to make the training more efficient, only a part of the complete trajectory will be sampled at a time. As it is discussed in the PPO paper Schulman et al. (2017), a truncated GAE is used for fixed-length trajectory segments, which is shown in Figure 1. Calculation of the GAE of an incomplete trajectory with length $T$ is represented as:

$$\hat{A}_T^{GAE(\gamma,\lambda,T)} = \delta_T^V = r_T + \gamma \cdot 0 - V(s_T) \tag{14}$$

$$\hat{A}_t^{GAE(\gamma,\lambda,T)} = \sum_{l=0}^{T-t} (\gamma\lambda)^l \delta_{t+l}^V \tag{15}$$

Following the GAE paper Schulman et al. (2015b), denote sum of $k$ of $\delta$ terms as $\hat{A}_t^{(k)}$. Let $\hat{A}_t^{(k)}$ be an estimator of the advantage function. When $k = 1$, $\hat{A}_t^{(1)} = \delta_t^V$, which has a large bias and low variance. The bias becomes smaller as $k$ becomes larger.

$$\hat{A}_t^{(k)} = \sum_{l=0}^{k-1} \gamma^l \delta_{t+l}^V = -V(s_t) + \gamma^k V(s_{k+t}) + \sum_{l=0}^{k-1} \gamma^l r_{t+l} \tag{16}$$

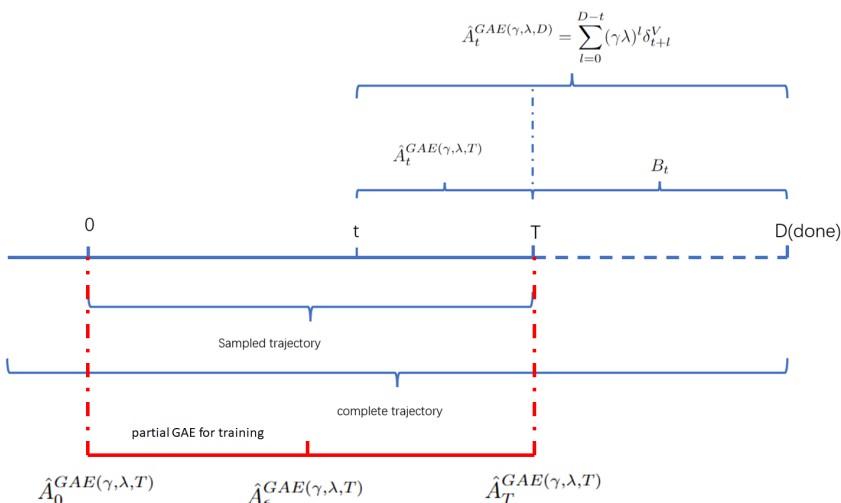

Figure 1: In practical application, considering parallel computing and avoiding episodes that are too long, a fixed sampling length is used. Since the last step of the sampling trajectory is not the termination of the complete trajectory, we have to use the truncated GAE as the estimator. For all of the calculated truncated GAE $\hat{A}_t^{GAE(\gamma,\lambda,T)}$, we propose to use $\hat{A}_t^{GAE(\gamma,\lambda,T)}$ for updating in case $t \leq \epsilon$ and discard $\hat{A}_t^{GAE(\gamma,\lambda,T)}$ in case $t > \epsilon$ as it shown in red part of this figure.

Since $\hat{A}^{GAE(\gamma,\lambda,T)}$ is the sum of exponentially-weighted $\hat{A}_t^{(k)}$, as $T - t$ increases, the number of cumulative terms increases, and the bias decreases.

$$\hat{A}_t^{GAE(\gamma,\lambda,T)} = (1 - \lambda) \sum_{k=1}^{T-t+1} \lambda^{k-1} \hat{A}_t^{(k)} \tag{17}$$

Consider two exceptional cases of $\hat{A}_t^{GAE(\gamma,\lambda,T)}$: $T = D$ and $t = T$. When $T = D$, $\hat{A}_t^{GAE(\gamma,\lambda,T)}$ is the same as $\hat{A}_t^{GAE(\gamma,\lambda,D)}$, and its bias is the minimum, that is, bias of $\hat{A}_t^{GAE(\gamma,\lambda,T)}$ will not be less than bias of $\hat{A}_t^{GAE(\gamma,\lambda,D)}$. For a specific trajectory, when $t = T$ and $T \neq D$, $\hat{A}_t^{GAE(\gamma,\lambda,T)}$ has the maximum bias, .

$$\hat{A}_t^{GAE(\gamma,\lambda,D)} = (1 - \lambda) \sum_{k=1}^{D-t+1} \lambda^{k-1} \hat{A}_t^{(k)} \tag{18}$$

$$\hat{A}_T^{GAE(\gamma,\lambda,T)} = r_t + \gamma V(s_{t+1}) - V(s_t) \tag{19}$$

Truncated GAE, similar to infinite $\hat{A}^{GAE(\gamma,\lambda)}$, balances between bias and variance with $\lambda$. $\hat{A}^{GAE(\gamma,1,T)}$ has low bias and large variance, while $\hat{A}^{GAE(\gamma,0,T)}$ has low variance and large bias. When $\lambda < 1$, with cost of bias, the variance is substantially reduced. Compared with infinite GAE, truncated GAE has larger bias and lower variance. Relatively, it is more important to reduce bias, which means for a specific trajectory as the step $t$ reduced, $\hat{A}_t^{GAE(\gamma,\lambda,T)}$ will become more instructive.

Compared with a GAE of a complete trajectory $\hat{A}_t^{GAE(\gamma,\lambda,D)}$, denote the difference between truncated $\hat{A}_t^{GAE(\gamma,\lambda,T)}$ and GAE $\hat{A}_t^{GAE(\gamma,\lambda,D)}$ as $B_t$:

$$B_t = \hat{A}_t^{GAE(\gamma,\lambda,D)} - \hat{A}_t^{GAE(\gamma,\lambda,T)} = \sum_{l=T-t}^{D-t} (\gamma\lambda)^l \delta_{t+l}^V \tag{20}$$

$B_T$ is a constant for a specific trajectory of length $T$. As shown in Equation 21, in the case of $0 < \gamma < 1$, $B_t$ decreases exponentially with step $t$. As $B_t$ reduced, the deviation between $\hat{A}_t^{GAE(\gamma,\lambda,T)}$

and $\hat{A}_t^{GAE(\gamma,\lambda,D)}$ is reduced. Since $\hat{A}_t^{GAE(\gamma,\lambda,T)}$ has larger bias than $\hat{A}_t^{GAE(\gamma,\lambda,D)}$, reduce $B_t$ can reduce the bias of $\hat{A}_t^{GAE(\gamma,\lambda,T)}$.

$$B_t = \sum_{l=0}^{D-T} (\gamma\lambda)^{l+T-t} \delta_{T+l}^V = (\gamma\lambda)^{T-t} B_T \tag{21}$$

In conclusion, in practical applications, due to the fixed length sampling trajectory, the calculation of GAE is truncated, which will lead to a large bias in the calculated GAE when $t$ is near the end of the trajectory. Instead, we propose to intercept a part of the GAE for use and drop the remainder of the trajectory with large bias. We propose a PPO algorithm with partial GAE as described in Algorithm 1, with partial coefficient $\epsilon$ and sample length $T$. In each iteration, each sampler collects $T$ samples and calculates $T$ truncated GAE. We take part of a GAE if $t > \epsilon$, the advantage estimates at time $t$ in the trajectory are discarded in the training. Then we construct the surrogate loss as Schulman et al. (2017) on these $N(T - \epsilon)$ data, and optimize the policy with minibatch Adam for $K$ epochs. For algorithm 1 according to Equation 17, as $T - t$ increases, the bias of $\hat{A}_t^{GAE(\gamma,\lambda,T)}$ decreases. As partial coefficient decreases, the bias of $\hat{A}_1, ..., \hat{A}_\epsilon$ decreases.

---

**Algorithm 1** PPO with partial GAE

---

    **for** iteration=1, 2, . . . **do**
      **for** actor=1, 2, . . . , N **do**
        Run policy $\pi_{\theta_{old}}$ in environment, get samples $(s_1, a_1, ..., s_T, a_T)$
        Compute advantage estimates $\hat{A}_1, ..., \hat{A}_T$
        **if** T is not done **then**
          only use $\hat{A}_1, ..., \hat{A}_\epsilon$ for updating
          keep $(s_\epsilon, a_\epsilon, ..., s_T, a_T)$ as $(s_1, a_1, ..., s_{T-\epsilon}, a_{T-\epsilon})$
        **else if** T is done **then**
          use $\hat{A}_1, ..., \hat{A}_T$ for updating
        **end if**
      **end for**
      **for** epoch K **do**
        Optimize surrogate $L$ wrt $\theta$, with minibatch size
        $\theta \leftarrow \theta_{old}$
      **end for**
    **end for**

---

## 4 EXPERIMENTS

We conducted a set of experiments to verify the effect of our proposed method, and conducted more detailed investigations:

- As we discussed above, can smaller partial coefficients reduce the bias of GAE for improved training?
- Can a larger sampling length reduce the bias of GAE to improve training?

We evaluated our method in MuJoCo OpenAI (2022) and $\mu$RTS Villar (2017). MuJoCo is a physical simulation environment. We mainly conducted experiments in Ant-v3 and complementary experiments in Halfcheetah-v3, Hopper-v3, Swimmer-v3, Walker2d-v3. $\mu$RTS is a simplified RTS game environment. Unlike MuJoCo, it has discrete states and discrete actions, and has a large number of game steps.

In the experiments with MuJoCo, we use Version 1.31 of MuJoCo (distributed with an MIT License). We use common values in applications to set Hyper-parameters. The discount factor $\gamma$ is 0.99, $\lambda$ for the GAE is 0.95, the clip coefficient of PPO is 0.2, the value coefficient is 1, the learning rate of the optimizer is 2.5e-4, the number of environments is 64, and the number of epochs is 2. To represent to policy, we use a three layer fully connected MLP (Multi-layer Perceptron) with a 64 unit hidden layer, and there are two additional noise layers after the MLP for exploration during training. The

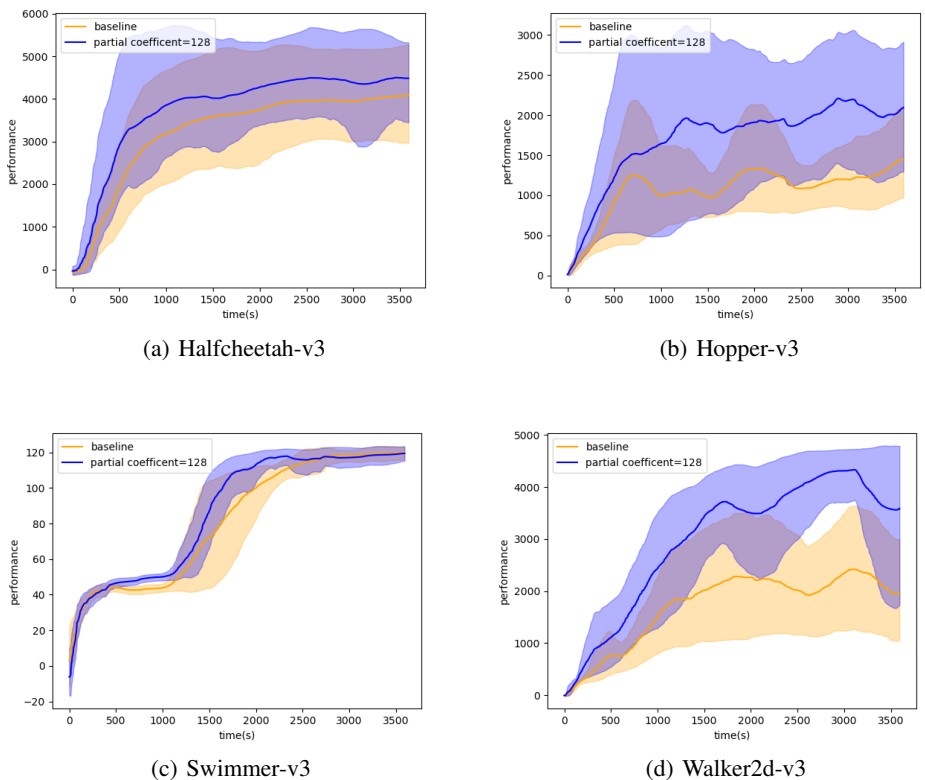

Figure 2: training curve in different MuJoCo environment

number of steps in a game of Ant-v3 is 1000, experiments were performed for sample length T from 128 to 1024 and coefficient $\epsilon$ from 64 to 512.

In the experiments with $\mu$RTS, we experimented in the $16 \times 16$ map against CoacAI which won the 2020 $\mu$RTS competition. In this experiment, the discount factor $\gamma$ is 0.99, $\lambda$ for the GAE is 0.95,the clip coefficient of PPO is 0.2, the entropy regularization coefficient is 0.01, the value coefficient is 1, the learning rate of the optimizer is 2.5e-4, the number of environments is 64, and the number of epochs is 2. To represent to policy, we use a two layer convolutional neural network connected to a three layer fully connected MLP with a 512 unit hidden layer.

In this paper, we use the training curve within a certain training time to evaluate the algorithm, rather than a fixed number of steps. This is because different algorithms and parameters will lead to differences in update time and training time. In practical applications, in addition to training effects, training time also should be considered. In the MuJoCo environment, we take the total reward of average 100 episodes episode after a certain training time as the performance score. In $\mu$RTS, we take the winning rate of recent 100 episodes after a certain training time as the evaluation index. For each set of variables, we used 3 random seeds for the experiment. We use an Intel(R) Xeon(R) CPU E5-2650 v4 @ 2.20GHz for our experiments.

### 4.1 WHAT IS THE EMPIRICAL EFFECT OF PARTIAL LENGTH AND SAMPLE LENGTH

As discussed above, the smaller the value of $t$ in truncated GAE $\hat{A}_t^{GAE(\gamma,\lambda,T)}$, the smaller the bias of the estimate, and the larger the variance. However, the variance of $\hat{A}_t^{GAE(\gamma,\lambda,T)}$ is always smaller than that of $\hat{A}_t^{GAE(\gamma,\lambda,D)}$, so theoretically, the partial coefficient should be as small as possible. In practical applications, small partial coefficients will lead to an increase in the number of GAE calculations, while a larger sampling length will lead to a longer sequence to be processed in a single GAE calculation, which will increase the calculation time.

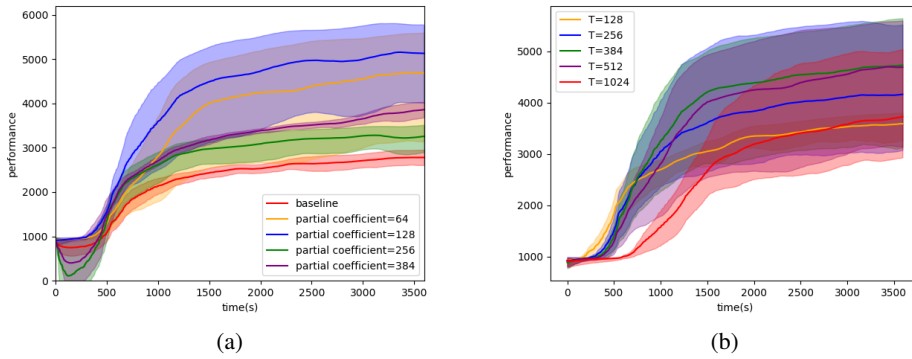

(a)                                      (b)

Figure 3: (a): training curve with different partial coefficient in Ant-v3, sample length $T = 512$. (b) training curve with different sample length in Ant-v3, partial coefficient $\epsilon = 64$.

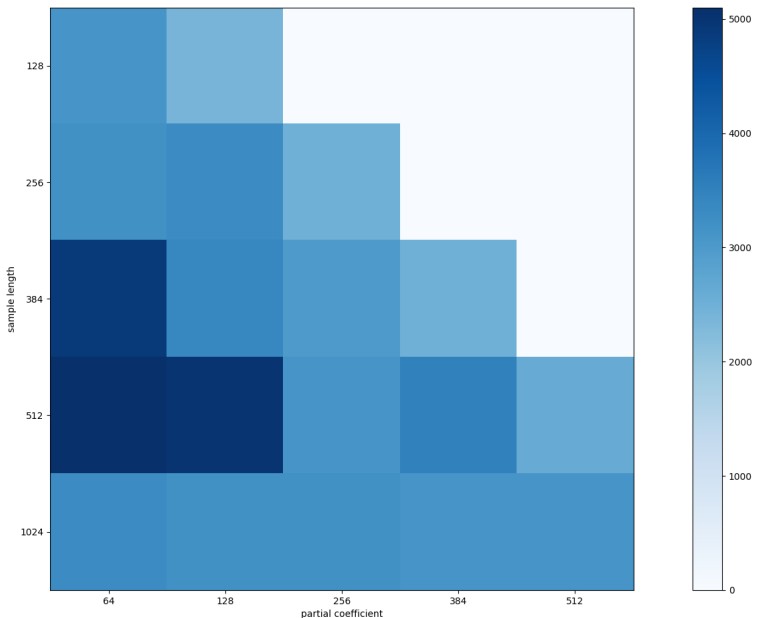

Figure 4: The performance after 1 hour training in Ant-v3. Since sample time $T$ is greater than or equal to partial coefficient $\epsilon$, the white part has no data. $T = \epsilon$ means not to discard any GAE, which is the baseline.

We compare common PPO as baseline with partial GAE PPO in Ant-v3, Halfcheetah-v3, Hopper-v3, Swimmer-v3, walker2d-v3, and we compare the effect of different partial coefficient and sample length in Ant-v3. Our implementation does not require the tricks of Engstrom et al. (2020) such as value clipping and an advantage normal.

The experimental results is shown in Figure 2 and Figure 3. In Figure 2 and Figure 3(a), using partial GAE can achieve better performance than the baseline, and improved performance can be obtained by using smaller partial coefficients. In Figure 3(b), using larger sample length improves performance. Figure 4 shows the performance after one hour training as $T$ and $\epsilon$ are varied. It can be seen that the highest performance score is with the partial coefficient $\epsilon \in [384, 512]$ and sample length $T \in [64, 128]$, in Ant-v3.

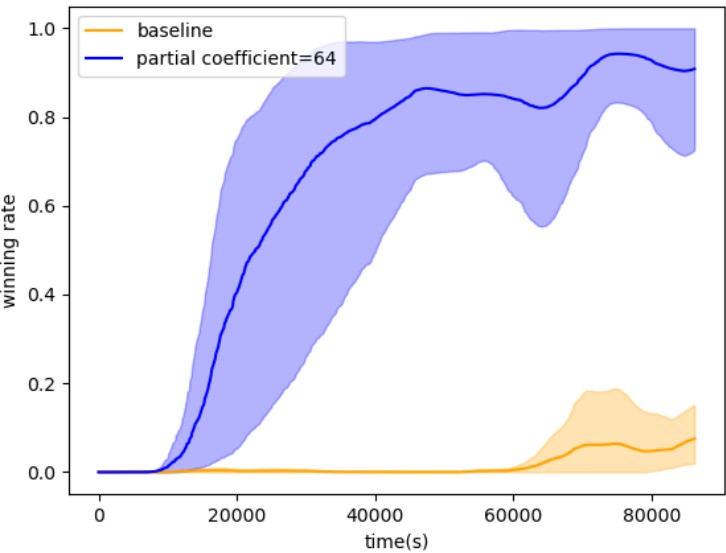

Figure 5: Winning rate during training in $\mu$RTS 16x16 map against CoacAI.

Note that when the partial coefficient is small or the sampling length is large, the performance does not improve by continuing to increase the partial coefficient or reduce the sampling length. As shown in Equation 21, when $(T - t)$ is large enough, $B_t$ will become very small, and the change brought by continuing to increase $(T - t)$ is very small. Although GAE significantly reduces variance by sacrificing bias, and the variance of truncated GAE is smaller than that of the complete trajectory, this does not mean that bias should be minimized while ignoring the impact of variance. In practical applications, the median of partial coefficient $\epsilon$ and sample length $T$ should be found to balance bias and variance. There is an intermediate value that makes the training effect the best, rather than maximizing $(T - \epsilon)$.

We conducted additional experiments in the 16x16 map of $\mu$RTS. This is a sparsely rewarded environment with long game steps. In this task, agents need to learn how to win in real-time strategy (RTS) games: collect resources, build units, and destroy enemy units and bases. The sample length $T$ is 512 in the experiments.

## 4.2 TRUNCATED ADVANTAGE ESTIMATOR VARIANCE INVESTIGATION

In the discussion in the above experiments, we supposed that blindly reducing the partial coefficient $\epsilon$ will lead to an increase in the variance of truncated GAE and make the training effect worse. In the MuJoCo environment, we recorded the 2000 GAE $\hat{A}_t$ to calculate standard deviation under different $t$ when the sampling length $T$ is 500, as shown in the Figure 6, red curve is the standard deviation of truncated GAE. As a whole, it can be seen from the figure that when $t$ is small, GAE has a larger standard deviation (or variance). However, unlike the previously mentioned theory, the variance does not completely decrease with the increase of $t$, especially at the end of a sampling sequence, the variance increases with the increase of $t$.

$$\hat{A}_t^{GAE(\gamma,\lambda,T)} = \hat{A}_t^r + \hat{A}_t^v \tag{22}$$

$$\hat{A}_t^r = \sum_{l=0}^{T-t} (\gamma\lambda)^l r_{t+l} \tag{23}$$

$$\hat{A}_t^v = \gamma(\gamma\lambda)^{T-t-1} V(s_T) - V(s_t) + \gamma(1-\lambda) \sum_{l=0}^{T-t-1} (\gamma\lambda)^l V(s_{t+l+1}) \tag{24}$$

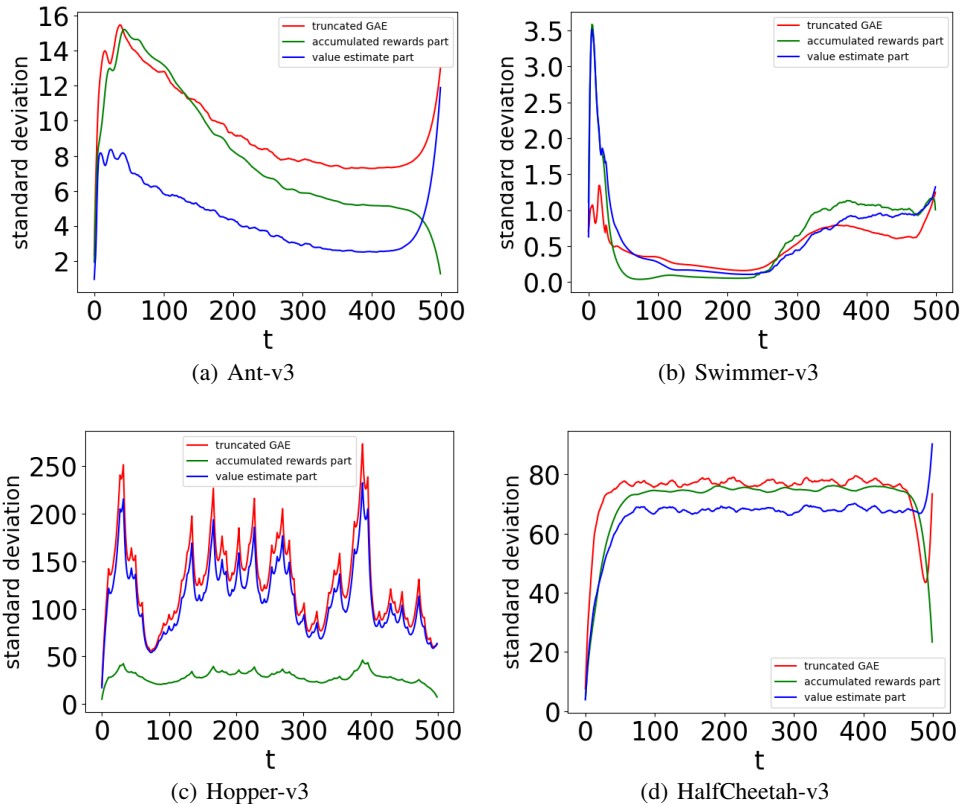

Figure 6: During Baseline training, the standard deviation under different sample time $t$. 2000 truncated GAE for a sample time $t$ after one million time steps are recorded to calculate standard deviation.

Write the truncated GAE in the form of the Eq. 22, which can be divided into two parts:accumulated reward part $\hat{A}_t^r$ and value estimate part $\hat{A}_t^v$. It can be seen from the Figure 2 that the variance is more affected by the value estimate part $\hat{A}_t^v$. And because of the uncertainty caused by the deviation of the value function fitting to the actual value, the variance of the truncated GAE does not only decrease with the increase of $t$. As shown in the Figure 6, empirically, smaller t will result in larger variance of truncated GAE. It is necessary to select an intermediate value of partial coefficient $\epsilon$ to balance the variance and bias of the truncated GAE to obtain better training effect.

## 5 CONCLUSION

How to estimate the value function is particularly important in policy gradient algorithms. GAE provides a method to balance the bias and variance of the estimation of the value function. However, in practical applications, truncated GAE is often used, and will lead to excessive bias of value estimation. We propose to use partial GAE in training to discard truncated GAE with excessive bias, which can reduce the bias and make value estimation more instructive.

We conducted experiments in MuJoCo environment. The experimental results show that using partial GAE will always achieve improved training results. For partial GAE related parameters, sampling length $T$ and partial coefficient $\epsilon$, although theoretically increasing $(T - \epsilon)$ can reduce bias, there is an intermediate value for the best training effect due to the influence of comprehensive variance. How to adjust sampling length $T$ and partial coefficient $\epsilon$ adaptively may be a direction for future research. We conducted additional experiments in $\mu$RTS. In this sparse environment with coefficient rewards and long game steps, partial GAE also performed well.

## 6 ETHICS STATEMENT

We note that our method does not introduce new potential societal harms as it is an improvement of value estimation; however it inherits any potential societal harms of deep reinforcement learning methods, which are well documented in Whittlestone et al. (2021). Note that the policy of an agent trained by deep reinforcement learning is highly dependent on its explored state and training environment, which causes the agent to perform unexpected actions when it encounters a state it has not seen before. It is not reliable to rely on the generalization of neural network to solve problems. It is necessary to consider how to deal with unexpected actions of agents, especially when DRL agents are applied to the real world.

## 7 REPRODUCIBILITY STATEMENT

Our experiments were repeated three times with different random seeds. We will upload our code in the supplemental materials for verification.

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
