# OpenReview forum: "Partial Advantage Estimator for Proximal Policy Optimization"
_ICLR.cc/2023/Conference — Submitted to ICLR 2023_

### Official Review · Reviewer_zF4x · 2022-10-23

**Confidence:** 4
**Correctness:** 2
**Technical Novelty And Significance:** 3
**Empirical Novelty And Significance:** 2
**Recommendation:** 3

**Clarity, Quality, Novelty And Reproducibility:**

**Clarity**

It is noteworthy that the paper is only 8 pages long including an ethics statement, unusually large figures and much white space (the page length limit is 9 pages). I don't think there is any paper accepted at a previous ICLR that does not fill the whole page limit. I also found the writing and figures to lack clarity (for example figure 1 is very difficult to understand). The notation could also be better, for example, typically $\epsilon$ is used for small real numbers, yet in the paper it represents an integer, so it can be confusing.

The introduction appeared more like a preliminaries section to me, as it started out with listing typical reinforcement learning technical definitions.

I found the flow of the text to be erratic with the topic suddenly changing, e.g., the sudden transition to discussing $q$ after equation 6, when the previous equations were about the TD method.

The figures in Figure 2 are very large and take a lot of space. The font sizes on the figure are too small. I believe these are png format figures, but it would be better to use a vector format such as pdf, so that the figures stay sharp when one zooms in.

**Quality**

It was not clear to me that the experimental results were solid. It seemed there was no comparison with existing implementations, so it is difficult to see whether their implementation is really performing well.

The errorbars on figure 2 are very large. Maybe increasing the number of samples would be good. Or perhaps using some other way to analyze the data would be good (e.g., see “Deep reinforcement learning at the edge of the statistical precipice”, NeurIPS2021)

It may be good to perform additional experiments other than only looking at reward learning curves. For example, the paper claims that the variance of $A_t$ will be the largest for low $t$. This could be verified experimentally by computing the variance. It would be even better if the bias is also estimated to show that large $t$ has a higher bias, but this may be harder to do.

The minimum tested value of T=128 seems large. Often in PPO the value is 5 or 10. How were these hyperparameters chosen?

Are the implementations good? It may be good to provide more explanation of this, e.g., show results of other implementations such as Stable Baselines. In $\mu\textup{RTS}$ are there any previous works that you can compare against? The performance looks good when compared to your implementation, but if you can explain that other researchers tried this task and their implementations perform worse than yours, it would provide much stronger evidence.

The method requires more memory than regular GAE methods. How does the method compare to a regular GAE with an increased batch size so that the amount of used memory is the same?

The experimental results on MuJoCo are reported against the wall-clock time. They explained why they did this, and it’s OK. However, it would be better to report both wall-clock time and environment steps. One of the two may be placed in the appendix.


**Novelty**

I have not seen this idea proposed before.

**Reproducibility**

I believe there were sufficient details to reproduce the work. The authors also provided their code.

**Strength And Weaknesses:**

Strengths
---------------
- The method appeared reasonable and simple to implement from a technical point of view.
- GAE is a very common technique, so the idea is relevant to many works.

Weaknesses
---------------
(See next section for justifications.)
- I found the clarity to be insufficient.
- It was not clear to me that the methods were well implemented as there did not appear to be a comparison with work or results produced by other researchers.
- The experimental results only looked at reward values. It would be good to find additional experimental evidence to support the claims, e.g. computing the variance of the advantage estimates (and the bias if possible).
- The work seems incomplete and not sufficiently thorough.
- The literature review is very short. For example "Asynchronous Methods for Deep Reinforcement Learning" Mnih 2016 that popularized the truncated rollout methodology was not cited.

**Summary Of The Paper:**

The standard GAE estimator uses a lambda weighted average of the k-step returns to estimate the advantage, i.e.,
it uses $\lambda(r_0 + \gamma V(s_1)) + \lambda^2 (r_0 + \gamma r_1 + \gamma^2 V(s_2) + ...)$ (in a normalized fashion).
In this standard method, the sequence continues until the terminal state or until $\infty$ so the sequence has the same bias-variance properties at any time-step. (The bias here comes from using a function approximator critic of $V$ that is not exact.) However, in practice we often use truncated rollouts of some short horizon T, shorter than the whole episode, so that the policy can be updated multiple times during a single episode. This allows learning faster as the policy is updated more often.

The current work points out that if we use truncated rollouts, then the bias-variance properties are different for small $t$ compared to large $t$ where $t<T$ is the time-step in the rollout. For example, when $t=T-1$ the advantage estimate is just $r_{T-1} + \gamma V(s_T) - V(s_{T-1})$, which relies more on the accuracy of the value critic compared to small $t$. In particular they mention that the bias might be large for large $t$, and the authors wish to reduce the bias.

To reduce the bias, they propose to do a partial GAE update, where they compute the advantage estimates for all $t<T$, but only update the policy for $t < H < T$. Then, they keep the data for $H \leq t \leq T$, gather more data, compute new GAE estimates, and update the policy parameters with the new data. This approach seems sensible as the gather data is not discarded. However, it seems there is a tradeoff of needing more memory. (The computation of the GAE estimates itself should be quite fast)

Experiments were performed on 4 MuJoCo tasks with 3 seeds each, where there was a trend towards the authors' partial GAE estimator improving the performance. They also performed experiments on $\mu RTS$ where the results showed that the new method quickly solves the task whereas the regular GAE does not.

**Summary Of The Review:**

I recommend rejecting the paper as I found it unclear and incomplete.

The idea of using a partial GAE estimate as presented in the paper appears like a sensible implementation trick, and I could see this being published if the experimental results are convincing and the paper is written well.

However, currently the quality of the paper seems far below the level I would expect to be published at ICLR (the length is only 8 pages, the writing is unclear, the experiments only look at reward values). I believe it requires much work on extending the experiments, and on improving the writing.

---

> ### Author Response · Authors · 2022-11-18
> **reply**
>
> In our device, a 1-hour mujoco experiment uses about 3 million or so samples. It can be compared with OpenAI's benchmark, and using our method can greatly improve the training. In Supplementary Material we provide code and more experimental data for verification.
>
> Added discussion of truncated GAEs to the paper.
>
> The number of steps in a game of mujoco is generally 1000. In different projects, the sampling length T ranges from 128 to 2048 are common parameters.

---

> > ### Comment · Reviewer_zF4x · 2022-11-20
> > **Thanks for the replay; however, I believe the work is not ready**
> >
> > Thank you for the response and the update to the paper.
> >
> > Indeed, as you mentioned the sampling lengths 128 to 2048 are common in implementations such as Stable Baselines, etc.
> > I was familiar with recent works using parallel simulators on a GPU (e.g., see Brax or Isaac Gym), where typically the good performance is achieved with much lower rollout lengths, which is why I made the comment about the testing with lower rollout lengths. I now believe, the tested rollout lengths are fine, so it's not a strong criticisms. However, you may want to try your algorithm on some of the recent fast simulators, as they would allow quickly running experiments with a larger amount of seeds, and in this case using shorter rollout lengths would be necessary.
> >
> > As most of my comments were not addressed, I will be keeping my score.
> > If using your method greatly improves the training on the OpenAI benchmark, I suggest running the experiments, and including the comparison in the paper.
> > The current experimental results are not that clear. The errorbars are too large, and it is not clear how the implementation compares to other works. Moreover, the presentation requires much polishing.
> > I encourage the authors to continue improving the experiments and the presentation of the results.

---

### Official Review · Reviewer_QPtY · 2022-10-26

**Confidence:** 3
**Correctness:** 4
**Technical Novelty And Significance:** 2
**Empirical Novelty And Significance:** 2
**Recommendation:** 5

**Clarity, Quality, Novelty And Reproducibility:**

Clear
Novelty is limited.

**Strength And Weaknesses:**

The work is very incremental and numerical results to demonstrate the performance are also very limited. From the paper, it seems like the performance is improved without paying any price which doesnt seem right to me

**Summary Of The Paper:**

This paper presents partial advantage estimation for PPO instead of using all the (truncated) advantage functions. PPO uses the lambda return method to estimate the advantage function. The proposed algorithm takes a partial coefficient \epsilon as input and discards trajectory for time steps greater than \epsilon, the advantage estimates are discarded in the training. Then the loss is computed on the remaining data.

**Summary Of The Review:**

Incremental improvement over PPO.

---

### Official Review · Reviewer_VSUL · 2022-10-27

**Confidence:** 4
**Correctness:** 2
**Technical Novelty And Significance:** 2
**Empirical Novelty And Significance:** 1
**Recommendation:** 3

**Clarity, Quality, Novelty And Reproducibility:**



The method of the paper is very clear. It's easy to reproduce the methods.

However, the paper does not contain enough solid evidence to support the claim of the paper.



**Details Of Ethics Concerns:**



**Strength And Weaknesses:**



Strength:

The proposed method is very simple and easy to implement.

It's also interesting to see that abandoning parts of the data near the end the trajectory can lead to better performance.



Weakness:

The writing of the paper is poor and casual. There are a lot of typos and errors. Many notations are undefined.

The paper seems to be a technical report of detail of the practical implementation. The contribution to the community is limited.

The experiments are done with only 3 seeds. It's hard to asses the results .



Several Questions:

I don't understand why  you assume that $V(s_t)=0$?

By Fig.3, It seems that the method is very sensitive to the hyperparameter of the partial coefficient $\epsilon$. How do you determine the value of this hyperparameter in practice?





Several typos or notations undefined:

eq.2: missing action $a$, $q_\pi(s,a)$, and please use big bracket.

Section 3 terminus: do you mean terminate?

eq. 11: why there is $V_{\theta_{t-1}}$ and $V_{\theta_{t+1}}$ ? By the way, do you define $V_{theta}$?

eq.12: Do you define $A_i$, $A_{mean}$, $A_{std}$

Figure 2: environment=>environments

**Summary Of The Paper:**

This paper proposed to abandon the samples near the end of the trajectory due to an estimation error in the GAE estimator. Experiments show the effectiveness of the method.

**Summary Of The Review:**



Generally, the paper is poorly written and contains a lot of typos and errors. The contribution of the paper is limited and it's hard to assess the effectiveness of the experimental results reported in the paper.

---

### Author Response · Authors · 2022-11-18
**rebuttal revision**

Modified in some details.
Added discussion of truncated GAEs to the paper.
Added more experimental data in Materials.

---

### Decision · Program_Chairs · 2023-01-20

**Decision:**

Reject

**Justification For Why Not Higher Score:**

The paper has major flaws and by no means ready for publication.

**Justification For Why Not Lower Score:**

N/A

**Metareview: Summary, Strengths And Weaknesses:**

The reviewers found the work incremental and without enough experimental support. They were also concerned about the lack of a good literature survey and proper comparison with other methods. Finally, the writing of the paper is not satisfactory with many typos, errors, and undefined notations. Although there might be some news ideas in the paper, its overall quality is low and requires more work (e.g., extending the experiments, proper comparison with other methods, and better writing) before can be properly assessed.